# Coastal Visibility Distance Estimation Using Dark Channel Prior and Distance Map Under Sea-Fog: Korean Peninsula Case

**DOI:** 10.3390/s19204432

**Published:** 2019-10-13

**Authors:** Tae Wuk Bae, Jin Hyun Han, Kuk Jin Kim, Young Taeg Kim

**Affiliations:** 1Daegu-Gyeongbuk Research Center, Electronics and Telecommunications Research Institute, Daegu 42994, Korea; 2Underwater Survey Technology (UST) 21, Incheon 21999, Korea; kjkim@ust21.co.kr; 3Division of Oceanographic Forecast, Korea Hydrographic and Oceanographic Agency, Busan 49111, Korea; kyt5824@korea.kr

**Keywords:** sea-fog, coastal visibility, dark channel prior, distance map, transmission

## Abstract

Commercial visibility sensors among meteorological sensors estimate the visibility distance based on transmission, backward scattering, and forward scattering principle. These optical visibility sensors yield comparatively accurate local visibility distance. However, it is still difficult to obtain comprehensive visibility information for a wide area, such as the coast or harbor due to the sensor structure using straightness and scattering properties of light. In this paper, we propose a novel visibility distance estimation method using dark channel prior (DCP) and distance map based on a camera image. The proposed method improves the local limit of optical visibility sensor and detects the visibility distance of a wide area more precisely. First, the dark channel for an input sea-fog image is calculated. The binary transmission image is obtained by applying a threshold to the estimated transmission from the dark channel. Then, the sum of the distance values of pixels, corresponding to the sea-fog boundary, is averaged, in order to derive the visibility distance. This paper also proposes a novel air-light and transmission estimation technique in order to extract the visibility distance for an abnormal sea-fog image, including any light source, such as sunlight, reflection light, and illumination light, etc. The estimated visibility distance was compared with optical visibility distance of an optical visibility sensor and their agreement was evaluated.

## 1. Introduction

Detection of various marine weather, such as sea-fog, typhoons, and high seas (or high winds) is being studied for the safety of civil and marine transit. Although, sea-fog has less direct influence or damage compared to typhoons or high seas, bad visibility due to sea-fog causes marine accidents, such as stranding or collision of a ship [1]. Marine scientific technology that predicts sea-fog occurrence and its movement is important, not only for the development of marine forecasting technology, but also for the reduction of marine accident and activation of nautical logistics for a national economy.

Due to the terrain conditions surrounded by sea on three sides in South Korea, there is much logistics transportation using container and passenger ships. The traffic volume of ships and the size of ports are becoming bigger and accidents also becoming more frequent as the scale grows. There are many causes of accidents, such as the ship’s defects and negligence of a person, but accidents caused by natural conditions, such as the wave and sea-fog account for a large part. For example, on 15 June 2015, a shipwreck accident occurred due to heavy sea-fog in Guryongpo, Pohang city of southern East coast of Korea. In the fog occurrence rate by sea area of South Korea in recent 10 years, the West coast has the largest amount of fog, corresponding to 55 % of total annual average number (108 cases) of fog observations, while the Jeju sea, the East coast, and the South coast have 19%, 14%, and 12%, respectively.

In the case of sea-fog occurrence of less than 1 km in South Korea, all passenger ships are controlled by Maritime Traffic Safety Act. On the west coast of South Korea, warm and humid air flows in a large amount from the Yellow-sea, due to the effect of westerly wind in the spring. In addition, the west coast has complex types of sea-fog due to the shallow depth of water, large tidal currents, complex coastline, and many small islands. Advection fog, which is a major sea-fog on the west coast, is a main cause of accidents, as the fog layer is thicker than land fog and occurs regardless of day and night [2]. In 2015, 106 cars collided with each other, due to the low visibility from sea-fog in Yeongjong Bridge.

Visibility means the maximum distance seen by the human eye. Fog is defined as a phenomenon in which very small droplets float in the atmosphere and the horizontal visibility is less than 1 km [3]. During the day, visibility is measured using Koschmieder’s law, which relates the minimum observable contrast between an appropriately large, black object against the horizon sky (called the contrast threshold) as a function of the atmospheric extinction coefficient and the distance to the target (e.g., the visibility) as the following [4],
(1)$$C_{T}=e^{-\beta V}$$
where $C_{T}$ is the observer’s contrast threshold, V denotes visibility in meters (or more specifically runway visual range for aviation purposes) and β is the atmospheric extinction coefficient, in m^−1^. In good visibility, the extinction coefficient is near to zero, and it increases as the visibility decreases. The extinction coefficient can also be named fog density [5]. Currently, commercial optical visibility sensors are manufactured based on transmission, backward-scatter, and forward-scatter principle [6]. Optical visibility sensors produce relatively accurate local visibility information. However, it is very expensive to deploy an optical visibility sensor to obtain comprehensive visibility information for large areas, such as the coast.

Recently, many studies have been conducted to improve visibility by removing fog area from camera images [7]. The haze removal methods, which is called dehazing (or defogging), have been introduced to restore haze pixels caused by absorption and scattering by atmospheric particles in haze. Due to the absorption and scattering by atmospheric particles in haze, outdoor images have poor visibility under inclement weather. Tan [8] proposed a method that takes into account the characteristic that a haze-free image has a higher contrast than a hazy image. The method enhances the visibility by maximizing the local contrast of the input hazy image, however, causes blocking artifacts around depth discontinuities. Fattal [9] proposed a method that infers the medium transmission by estimating the albedo of the scene. It is based on the assumption that the transmission and surface shading are locally uncorrelated, so defogging is done by estimating the scene albedo and then inferring the transmission map. He et al. [10] proposed a novel prior-dark channel prior (DCP) by observing the property of haze-free outdoor images. The DCP is based on the property of dark pixels, which have a very low intensity in at least one color channel, except for the sky region. Owing to its effectiveness in dehazing, the majority of recent dehazing techniques [11,12,13,14,15,16,17,18] have adopted the DCP.

Recently, DCP applications have been developed to reconstruct underwater scenes or hazy images [19,20,21,22]. Although, its direct application to underwater images as-is is not suitable and creates inconsistent results. Akkaynak et al [19,20] modified the DCP method for underwater images and proposed a method for removing water from these images by estimating the range-dependent attenuation coefficient. Ren et al. [21] presented a gated fusion network using three derived input images from an original image and symmetric encoder and decoder for single image dehazing. Zhang et al. [22] proposed a unified dehazing network for estimating the transmission map precisely. These introduced methods can be used in vehicles and ships to ensure visibility in coastal areas. These DCP-based findings can also be applied to quantitatively estimate the visibility distance in coastal areas, where sea-fog occurs frequently.

This paper introduces a visibility distance detection algorithm using DCP and distance map for improving the local limit of the optical visibility system and detecting the visibility distance of the wide area more accurately. First, the dark channel image for input sea-fog image is formed. The binary transmission image is obtained by applying a threshold to the estimated transmission obtained by using the dark channel and the air-light. Then fog boundary for the input sea-fog image is extracted by simple image processing. Then, the sum of distance values of the pixels corresponding to the extracted sea-fog boundary is averaged to derive the visibility. This paper also proposes a new air-light and transmission estimation method for abnormal sea-fog images including sunlight and illumination light. The estimated visibility distance was compared with the visibility distance of optical visibility sensor, and the similarity was evaluated.

## 2. Materials and Methods

### 2.1. Types of Sea-Fog

Sea-fog is a meteorological phenomenon that causes a disorder similar to fog. Fog occurs in various forms, depending on humidity, temperature, wind, type, and the amount of condensation nucleus [23]. Fog is usually divided into cooling-fog and evaporation-fog, depending on the generation method. The cooling-fog occurs when warm air layer meets the sea surface and air temperature falls below the dew point, and the evaporation-fog occurs when water vapor evaporates. The cooling-fog is divided into a radiation-fog formed by the cooling of the surface, an advection-fog that occurs when warm, wet air is saturated after it has moved over a water surface, and an upslope-fog where wet air ascends and condenses along a high terrain. The evaporation fog is divided into a frontal fog, which occurs at low rainfall near the warm front, and a steam fog, which occurs when cold air moves on warm water.

Sea-fog occurs when warm air saturates close to the cold sea surface. Sea-fog may enter inland from the ground, depending on the wind system at the altitude of 925 hPa, long-lasting when created simultaneously with inland radiation-fog in a stable atmosphere, and worsening coastal and inland visibility. Sea-fog is also located at the center of high pressure, occurs when wind is strong or there is lower layer cooling, due to advection to relatively cool water [24]. Thus, sea-fog directly affects the coastal area, causing low visibility due to fog, and ascending as it moves inland, creating a stratiform cloud, a low-level cloud.

### 2.2. Image Formation Model and DCP

The haze imaging equation [9] for a hazy image in the atmosphere is given as follows,
(2)I(x)=J(x)t(x)+A(1−t(x))
where x represents the pixel location, I is the foggy image, J is the defogged image (i.e., scene radiance). While, $t\left(x\right)=e^{-\beta d\left(x\right)}$ is the transmission map, which shows how much of the light in the scene radiance is not scattered by fog or haze particles and reaches the camera, and β is the scattering coefficient of the atmosphere, is the scene depth, distance from the object to the camera. A is the atmospheric light intensity, which is assumed to be the same for every pixel. The J(x)t(x) term represents the direct attenuation, i.e., how much of the scene information reaches the camera without scattering. The A(1−t(x)) term represents air-light, i.e., how much the atmospheric light contributes to the foggy image. In clear weather conditions, we have β≈0, and thus I≈J. However, β becomes non-negligible for hazy images. The direct attenuation decreases as the scene depth increases. In contrast, the air-light increases as the scene depth increases.

He et al. [10] performed an empirical investigation of the characteristic of haze-free outdoor images. They found that there are dark pixels whose intensity values are very close to zero for at least one color channel within an image patch. Based on this observation, a dark channel is defined as follows,
(3)$$J^{dark}\left(x\right)=\underset{y\in \Omega \left(x\right)}{\min}\left(\underset{c\in \left(r,g,b\right)}{\min}J^{c}\left(y\right)\right)$$
where $J^{c}$ is an intensity for a color channel c∈(r,g,b) of the RGB image and Ω(x) is a local patch centered at pixel x. According to Equation (3), the minimum value among the three color channels and all pixels in Ω(x) is chosen as the dark channel $J^{dark}\left(x\right)$. A sea-fog image is brighter than the corresponding clear image due to the added air-light. The dark channel for a sea-fog image has a higher intensity. Therefore, the intensity of the dark channel represents the fog density.

### 2.3. Proposed Visibility Distance Detection Method

The block-diagram of the proposed visibility distance detection method is shown in Figure 1. First, by using the dark channel image for an input image, it is distinguished whether the input image is a normal sea-fog image or an abnormal sea-fog image, including any light source. For the normal sea-fog image, the transmission map is derived from the dark channel of the input image, then the sea-fog boundary is detected by applying a threshold to the transmission map. Finally, the visibility distance is obtained by averaging the distance values of pixels corresponding to the detected sea-fog boundary. In the conditional statement determining a normal or an abnormal sea-fog image, if the number of pixels with dark channels, valued more than 0.9, are more than 0.01% of the total number of pixels of the input image, the input image is considered as an abnormal sea-fog image. For the abnormal sea-fog image, using modified transmission and light source detection technique proposed in this paper, the sea-fog boundary is detected. Then, the visibility distance is obtained as for the normal sea-fog image.

#### 2.3.1. Overview for Obtaining Transmission in Image

The dark channel can be given using the image formation model as the following,
(4)$$I^{dark}\left(x\right)=J^{dark}\left(x\right)t\left(x\right)+A^{c}\left(1-t\left(x\right)\right)$$
where $I^{dark}\left(x\right)$ represents the dark channel for an input sea-fog image and $J^{dark}\left(x\right)$ means the dark channel for the clear image. While, $A^{c}$ represents the color channel of air-light in the input sea-fog image. As most of the intensity values of the dark channel for the clear image are zero, Equation (4) can be expressed as the following:(5)$$I^{dark}\left(x\right)\approx A^{c}\left(1-t\left(x\right)\right).$$

Consequently, the transmission map can be estimated by transforming Equation (5) as the following:(6)$$t\left(x\right)=1-I^{dark}\left(x\right)/A^{c}.$$

The t(x) for the sky area in a sea-fog image is nearly zero as that area has an infinite distance value, which means that the color value of the sky area is very similar to the intensity of the air-light.

#### 2.3.2. Visibility Sensor and CCTV Information for Each Port

The three major ports in South Korea are Busan port, Busan New port and Incheon port. The visibility sensor equipped in each port is PWA22 of Vaisala as shown in Figure 2a. The PWD22 has a range of 10–20 km and is a visibility sensor with a forward scattering method. As mentioned in the introduction, the optical visibility sensor yields a relatively accurate local visibility distance value, but has difficulty in calculating the broad visibility distance for a wide coastal area. Nevertheless, optical visibility distance of the optical visibility sensor is used as a reference for the performance verification of the CCTV-based visibility distance detection method, as proposed in this paper.

The CCTV equipped for sea-fog surveillance is HIKVISION’s DS-2DF8836IV-AELW, an outdoor PTZ dome camera with 1/1.9” HD CMOS sensor as shown in Figure 2b. Figure 3 shows each port, where CCTV is equipped and its scene in a streaming video. Each port has an object (island) for more accurate distance estimation in the image, which are Jodo for Busan port, Yeongjongdo for Incheon port, and Yeondo for Busan New port, respectively. In order to estimate the visibility distance in the proposed method, distance values for all pixels in the image should be calculated. In this paper, it is called the distance map, which is obtained by interpolating the actual distance value to each island and some actual distances obtained from Google satellite map. The detailed description is given in Section 2.3.3. As a result, the visibility distance is obtained by averaging the distance values corresponding to the detected sea-fog boundary pixels.

#### 2.3.3. Distance Map

Figure 3 also shows the distance map for each port used to calculate the visibility distance in the proposed visibility distance detection method. The distance map is generated by interpolating with actual distance from each CCTV position to each island and some obtained actual distances (the blue dot in the Figure of the fourth column) for the vertical pixels of the images. In the Figure, the second column shows actual distance from each CCTV position to each island. The third, and fourth column, respectively show the estimated distance map and the distance values for vertical pixels in the estimated distance map. In the Figures in the fourth column, the blue dot represents actual distance values obtained from Google satellite map, and the red line represents the interpolated distance values. Since the proposed method estimates the visibility distance using distance values, which correspond to the detected sea-fog boundary, the accuracy of the distance map affects the visibility estimation. The distance map used in the experiments may include a small distance error due to the interpolation, as the entire distance values were interpolated by some actual distance values. However, we confirmed that it works well in the visibility distance estimation experiment.

#### 2.3.4. Proposed Visibility Distance Detection Method

• Case I: Normal Sea-fog Image

The Block-diagram for calculating the visibility distance for a normal sea-fog image is shown in Figure 1. First, the dark channel for an input sea-fog image is created by Equation (3), then the transmission map is estimated by Equation (6). The binary transmission image is given by applying a threshold to the estimated transmission map, then the sea-fog boundary is detected by applying the binary pattern to the binary transmission image. Finally, the visibility distance is obtained by averaging the distance values of pixels corresponding to the detected sea-fog boundary.

The binary transmission image B(x) is given by applying the threshold $T_{fb}$ to the transmission image for distinguishing sea-fog pixel and background pixel as the following,
(7)$$B\left(x\right)=\{\begin{array}{cc} -1\;\left(i.e.\;\mathrm{sea}-\mathrm{fog}\;\mathrm{pixel}\right) & \mathrm{if}\;T\left(x\right)<T_{fb}\\ 1\;\left(i.e.\;\mathrm{background}\;\mathrm{pixel}\right) & \mathrm{else} \end{array}$$
where “−1” represents a sea-fog pixel with low transmission value and “1” represents a background pixel with high transmission value. The pattern mask $M_{k}\left(-p\leq k\leq p,\;p=3\right)$ for detecting the sea-fog boundary is given as the following:(8)$$M_{k}=\{\begin{array}{cc} -1 & \mathrm{for}\;-3\leq k\leq 0\\ 1 & \mathrm{for}\;1\leq k\leq 3 \end{array}.$$

Finally, the sea-fog boundary image F(x) is obtained by assigning “1” to pixel positions having the same pattern as the pattern mask and assigning “0” to other pixel positions in the binary transmission image. Mathematically, it is equivalent to satisfying the following condition,
(9)$$F\left(x\right)=\{\begin{array}{cc} 1 & \mathrm{if}\;\overset{-1}{\underset{i=-p}{\sum }}B_{i}\left(x\right)=\overset{p}{\underset{i=1}{\sum }}B_{i}\left(x\right)\mathrm{and}\;B_{1}\left(x\right)=\left(-1\right)B_{0}\left(x\right)\\ 0 & \mathrm{else} \end{array}$$
where 1 and 0 denote the sea-fog boundary pixel, and background pixel, respectively. $B_{i}\left(x\right)$ means the column matrix corresponding to (−i~+i) position centered on *x* location.

• Case II: Abnormal Sea-fog Image Including Light Source

For an abnormal sea-fog image, including a light source, such as sunlight, reflection light, and illumination light etc., the visibility distance may not be correctly calculated through basic transmission and air-light estimation method, used in the traditional DCP technique [10]. Since pixel values of a light source region are very high, it may be incorrectly detected as a sea-fog region. Because of this reason, this paper presents a novel air-light and transmission estimation method to calculate the visibility distance correctly in an abnormal sea-fog image.

In the conventional DCP technique [10], air-light is estimated by the average RGB value of the pixels, corresponding to the top 1% of pixel values of the dark channel. Figure 4 shows the proposed air-light estimation method for an abnormal sea-fog image. In the case of the conventional air-light estimation method, the sunlight region or the reflection light region is wrongly used for air-light estimation, since the top 1% in the dark channel value correspond to the regions. In order to address these drawbacks, a novel air-light estimation method using variance-mean of the dark channel is introduced.

First, an input image with M×N size is divided into 10 × 10 blocks with (M/10)×(N/10) size. The variance-mean map with 10×10 size for the dark channel is made by dividing the block-variance by the block-mean for each block, then the top five blocks are selected. Among the original pixels, corresponding to the selected blocks, the pixels with high dark channel value of 1% of the total image size, are used for air-light estimation. This process makes the sky region around the light source more prominent than the light source region at air-light estimation. The light source region has a high dark channel value but has a low variance, due to its flat characteristics. On the other hand, the dark channel values of the sky region around the light source are smaller than the light source region but has a high variance because there is a complex region. As a result, it can be seen that air-light is estimated from the pixels in the sky region around the light source.

Figure 5 shows a novel transmission estimation method for an abnormal sea-fog image. In case a light source exists, the conventional transmission estimation method in the conventional DCP [10] shows the lowest transmission in the light source region. As a result, the sunlight region or the reflection light region are misinterpreted as a sea-fog region. In order to address these drawbacks, pixel regions (sunlight or reflection light region), greater than 1.5 times the estimated air-light by the novel air-light estimation method, are classified as an abnormal map, then excluded in the transmission estimation. Thereafter, this abnormal map, the pixel positions corresponding to the light source, such as the sunlight region and the reflection light region, are excluded from the transmission estimation (i.e., the transmission of the abnormal map is set to 1). As shown in the Figure 5, the pixel positions, corresponding to the light source, are excluded from the transmission estimation, even though the regions have very high, dark channel values. As a result, it can be seen that the fog regions around the sunlight have the lowest transmission.

## 3. Results

### 3.1. Analysis of Case I (Normal Sea-Fog Image)

#### 3.1.1. Comparison of Estimated and Optical Visibility Distance

This section compares the estimated visibility distance by the proposed method and the optical visibility distance measured by the optical visibility sensor. The image sequences, used for the experiment, are 100 frames for Busan port and Incheon port and 52 frames for Busan New port with 704 × 576 resolution, and are obtained from the CCTV equipped in each port. The image sequences were taken during the daytime when there was sea-fog. The visibility distance, estimated by the proposed method, represents the visibility distance of the coastal area photographed by the CCTV camera. Whereas, the optical visibility distance, measured by the optical visibility sensor, represents the local visibility distance. Therefore, there may be some differences between these visibility distances.

Figure 6 shows the dark channel, transmission, binary transmission, and sea-fog boundary detection results for the input sea-fog images taken at each port. Figure 6a shows the sea-fog boundary detection results for the sea-fog images at Pusan port with $T_{fb}=0.4$ for the short distance (thick) sea-fog and $T_{fb}=0.35$ for the long distance (weak) sea-fog. In the transmission map obtained through the dark channel, the non-sea-fog region has higher (brighter) transmission, while the sea-fog region has lower (darker) transmission value. The binary transmission classifies the sea-fog region and the non-sea-fog region, and the estimated visibility distance, was calculated as an average of distance values of pixels corresponding to the sea-fog boundary. In the case of the long distance sea-fog, the sea-fog boundary pixels were mainly detected between Jodo and horizon. On the other hand, in the case of the short distance sea-fog, the sea-fog boundary pixels were detected at the actual sea-fog boundary. The difference between the estimated visibility distance and the optical visibility distance is 65–277 m for the short distance sea-fog, and 825–2498 m for the long distance sea-fog, respectively. It is experimentally demonstrated that the estimated visibility distance is very similar to the optical visibility distance.

Figure 6b shows the sea-fog boundary detection results for the sea-fog images at Incheon port with $T_{fb}=0.38$ for the short distance sea-fog and $T_{fb}=0.4$ for the long distance sea-fog. In the case of the long distance sea-fog, the sea-fog boundary pixels were mainly detected between Yeongjongdo and horizon. On the other hand, in the case of the short distance sea-fog, the sea boundary pixels are detected at the actual sea-fog boundary. The difference between the estimated visibility distance and the optical visibility distance is 249–531 m for the short distance sea-fog and 819–2663 m for the long distance sea-fog. The difference of visibility distance for the short distance sea-fog at Incheon port is larger than that of Busan port. The reason is that the distance value per pixel in the image of Incheon port is larger than that of Busan port due to the low CCTV height. Based on each equipped CCTV location, Yeongjongdo is farther than Jodo. For the short distance sea-fog, errors may occur because the outline of the tree can be mistaken for sea-fog boundary.

Figure 6c shows the sea-fog boundary detection results for the sea-fog images at Busan New port with $T_{fb}=0.47$ for the short distance sea-fog and $T_{fb}=0.52$ for the long distance sea-fog. In the case of Busan New port, the distance value per pixel is smaller than that of Incheon Port, due to the high CCTV height. The accuracy of the estimated visibility distance is higher compared to other ports. The difference between the estimated visibility distance and the optical visibility distance was 4–408 m for the short distance sea-fog, and 30–1171 m for the long distance sea-fog. However, in the case of Busan New port, the outline of the crane may be erroneously detected as the sea-fog boundary, which may cause inaccuracy.

#### 3.1.2. Setup of Threshold

Figure 7 shows the comparison of the estimated visibility distance and the optical visibility distance for (a) Busan port, (b) Incheon port, and (c) Busan New port according to various thresholds applied to the transmission map. It can be seen that the difference changes by varying the threshold. The long distance and short distance sea-fog mean the optical visibility distance within 20 km, and 5 km, respectively. If the threshold is small, even a very weak sea-fog region is detected (i.e. sensitive to sea-fog). On the other hand, if the threshold is large, only a very thick sea-fog region is detected (i.e., insensitive to sea-fog). The reason for this difference is that the optical visibility sensor only calculates the visibility distance for a specific axis in a three-dimensional space. Whereas, the proposed method yields the visibility distance for actual space. As shown in the Figure, the difference of the visibility distance in the long distance sea-fog is larger than that for the short distance sea-fog. This is because the distance value corresponding to one-pixel increases as the distance increases.

In case of Busan port, the mean of the absolute value of the differences was 3201 m, 2825 m, 3123 m, 3626 m, 4061 m, and 4442 m for the long distance sea-fog and 1384 m, 961 m, 611 m, 559 m, 612 m, and 687 m for the short distance sea-fog with the thresholds ($T_{fb}=0.25,\;0.3,\;0.35,\;0.4,\;0.45\;\mathrm{and}\;0.5$). It can be seen that $T_{fb}=0.3$ for the long distance sea-fog and $T_{fb}=0.4$ for the short distance sea-fog yield the smallest difference.

In the case of the Incheon port, the mean of the absolute value of the differences was 2364 m, 1890 m, 2143 m, 2476 m, and 3522 m for the long distance sea-fog and 2152 m, 1237 m, 864 m, 876 m, and 1020 m for the short distance sea-fog with the thresholds ($T_{fb}=0.35,\;0.375,\;0.4,\;0.425\;\mathrm{and}\;0.45$). It can be seen that the difference is the smallest when $T_{fb}=0.375$ for the long distance sea-fog and $T_{fb}=0.4$ for the short distance sea-fog. The height of the CCTV equipped at Incheon port is lower than that of Busan port, so the distance value corresponding to one pixel is larger than Busan port. As a result, it can be seen that the difference for the short distance sea-fog at Incheon port is larger than Busan port. However, the difference for the long distance sea-fog was smaller compared to the Busan port.

In the case of Busan New Port, the optical visibility distance was within 5 km. The mean of the absolute value of the difference was 2815 m, 1807 m, 1307 m, 984 m, 777 m, and 674 m for the short distance sea-fog with the thresholds ($T_{fb}=0.25,\;0.3,\;0.35,\;0.4,0.45,\;\mathrm{and}\;0.5$). It can be seen that the difference is the smallest when $T_{fb}=0.5$ for the sea-fog. Since the CCTV height of Busan New Port is higher than that of Incheon port, the difference error for the short distance sea-fog is less than that of Incheon port.

#### 3.1.3. Correlation Coefficient

We performed the principal component analysis (PCA) to analyze the correlation between the estimated visibility distance and the optical visibility distance. Figure 8 shows the correlation coefficient results for Busan port according to various thresholds applied to the transmission map. The correlation coefficients for the thresholds ($T_{fb}=0.25,\;0.3,\;0.35,\;0.4,\;0.45\;\mathrm{and}\;0.5$) were 0.776, 0.841, 0.877, 0.917, 0.909, and 0.901 for the long distance sea-fog and 0.784, 0.824, 0.867, 0.885, 0.879, and 0.881 for the short distance sea-fog. In the case of Busan port, we can see that the correlation coefficient is the highest for all sea-fogs when $T_{fb}=0.4$.

Figure 9 shows the correlation coefficient results for Incheon port. The correlation coefficients for the thresholds ($T_{fb}=0.35,\;0.375,\;0.4,\;0.425,\;\mathrm{and}\;0.45$) were 0.891, 0.920, 0.943, 0.936, and 0.908 for the long distance sea-fog, and 0.806, 0.758, 0.681, 0.615, and 0.611 for the short distance sea-fog. In the case of Incheon port, the largest correlation coefficient was obtained when $T_{fb}=0.4$ and $T_{fb}=0.35$ for long distance and short distance sea-fog, respectively.

Figure 10 shows the correlation coefficient results for Busan New port. The correlation coefficients for the thresholds ($T_{fb}=0.25,\;0.3,\;0.35,\;0.4,0.45,\;\mathrm{and}\;0.5$) were 0.596, 0.744, 0.769, 0.815, 0.832, and 0.807. In the case of Busan New port, we can see that the correlation coefficient is the highest when $T_{fb}=0.45$.

### 3.2. Analysis of Case II (Abnormal Sea-fog Image including Light Source)

Figure 11 shows the sea-fog boundary and visibility estimation detection for abnormal sea-fog images, including the sunlight and reflection light (Incheon port), as well as the illumination light (Busan New port). The proposed method prevents air-light from being detected in the light source region. By replacing the transmission value of the light source region with “1”, the proposed method eliminates the error where the light source region is incorrectly detected as sea-fog region. We can confirm that the proposed method can detect the sea-fog region even if any light source exists.

The input image of Incheon port has the sunlight region and the reflection region with very high pixel values (almost 255). The regions were misinterpreted as a thick sea-fog region because the estimated transmission of the regions are very low. In the proposed method, the regions are classified as the abnormal map and are not mistaken for a sea-fog region. The proposed method detects only the region around the light source as the sea-fog boundary. For the optical visibility distance of 20 km, the proposed method estimates the visibility distance of 16.96 km, while the conventional method predicts 9.41 km. The threshold value for distinguishing between the sea-fog and the background pixels, used in this experiment, was $T_{fb}=0.3$. Also, in case of the input image of Pusan new port with the illumination light, the proposed method respectively estimate the visibility distance of 2.2 km, and 1.6 km, for optical visibility distance of 0.58 and 0.36 km. There was some error, but the proposed method greatly reduces the estimation error compared with the conventional DCP method.

## 4. Discussion

To replace the existing optical visibility sensor and obtain a comprehensive visibility distance information in the coastal area, a CCTV-based visibility distance estimation method is proposed. Through the experimental results, it is deduced that the proposed method can detect reliable result for a sea-fog within 5 km when $T_{fb}=0.4$. In addition, it was confirmed that the visibility distance, estimated by the proposed method, can better represent the sea-fog distribution in the wide coastal area compared with the optical visibility distance.

To improve the accuracy of the visibility distance estimation, the following CCTV situations should be considered. First, the objects or structures in CCTV images should be excluded. If an object is completely covered by a sea-fog, the boundary of the object can be misinterpreted as a sea-fog boundary. Therefore, CCTV should be installed in the direction of no object or structure, if possible. Second, the height of the CCTV camera should be considered. The CCTVs of each port were installed at different heights due to the surrounding terrain. This causes the distance value corresponding to one pixel to be different. The larger the distance value corresponding to one pixel, the greater the error in the visibility distance estimation. So CCTV should be installed as high as possible. Third, atmospheric scattering occurs around any light source. Even if the light source region is removed through the proposed method, pixels whose pixel value has changed, due to atmospheric scattering, still exist in the image and affect the visibility distance estimation. If CCTV is installed in consideration of these precautions, it will be possible to calculate more accurate visibility distance estimation under given conditions.

Additionally, atmospheric correction techniques for distinguishing and water contributions have been researched in the ocean color remote sensing [25,26]. The studies are related to the absorption or reflection properties of water in the infrared wavelength band, modeling water properties, and ocean inversion using neural networks. These studies can also provide global atmospheric and ocean segmentation information in coastal areas. Therefore, if the proposed technique is combined with the atmospheric correction technique in the future, more reliable visibility distance can be estimated.

## 5. Conclusions

In this paper, we propose a CCTV-based visibility distance estimation method that can be used in wide coastal areas. The performance of the proposed method was successfully evaluated for three representative ports in South Korea. Although, the accuracy of the optical visibility sensor is guaranteed, it is difficult to comprehensively estimate the visibility of wide areas, due to its locality. The proposed method estimates the visibility distance by applying the transmission information based on the image formation model. The existing defogging algorithms works well only in weak sea-fog situations, where there is no light source. The proposed method estimates the visibility distance relatively accurately, through the novel air-light and transmission estimation, even under the circumstances where the existing defogging methods do not estimate accurately. Through this study, it was confirmed that CCTV-based signal processing could calculate the fog area and the visibility distance.

## Figures and Tables

**Figure 1 sensors-19-04432-f001:**
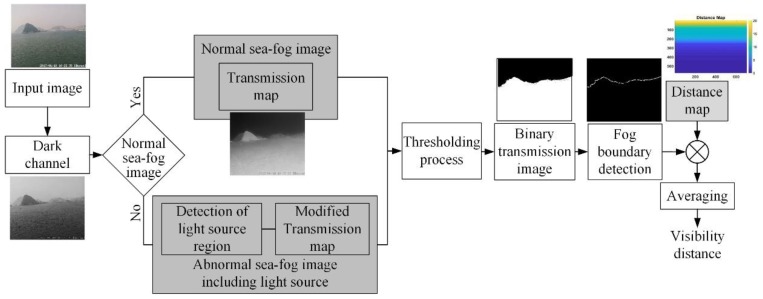
Block-diagram of proposed visibility distance detection method.

**Figure 2 sensors-19-04432-f002:**
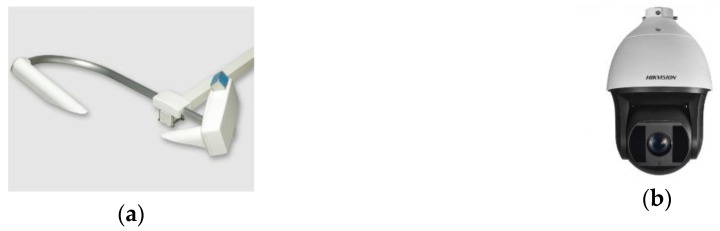
(**a**) Visibility sensor and (**b**) CCTV equipped in each port.

**Figure 3 sensors-19-04432-f003:**
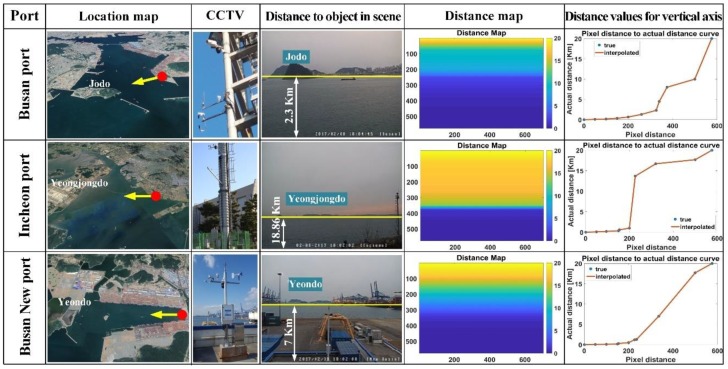
Island information, CCTV, and distance map for each port.

**Figure 4 sensors-19-04432-f004:**
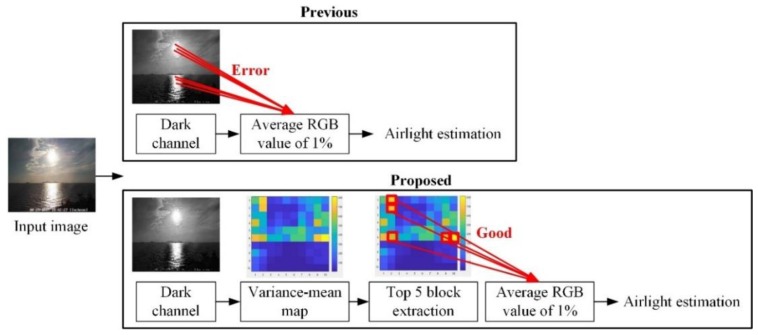
Proposed air-light estimation method for abnormal sea-fog image.

**Figure 5 sensors-19-04432-f005:**
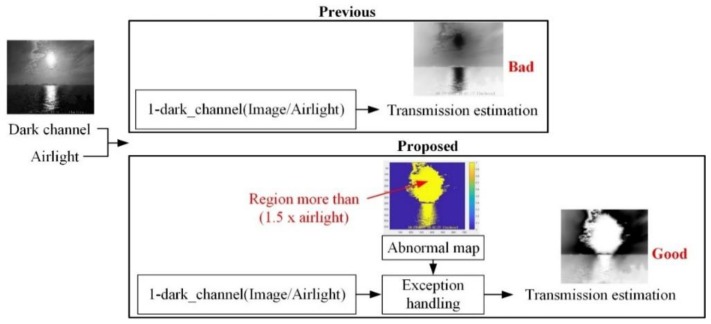
Proposed transmission estimation method for abnormal sea-fog image.

**Figure 6 sensors-19-04432-f006:**
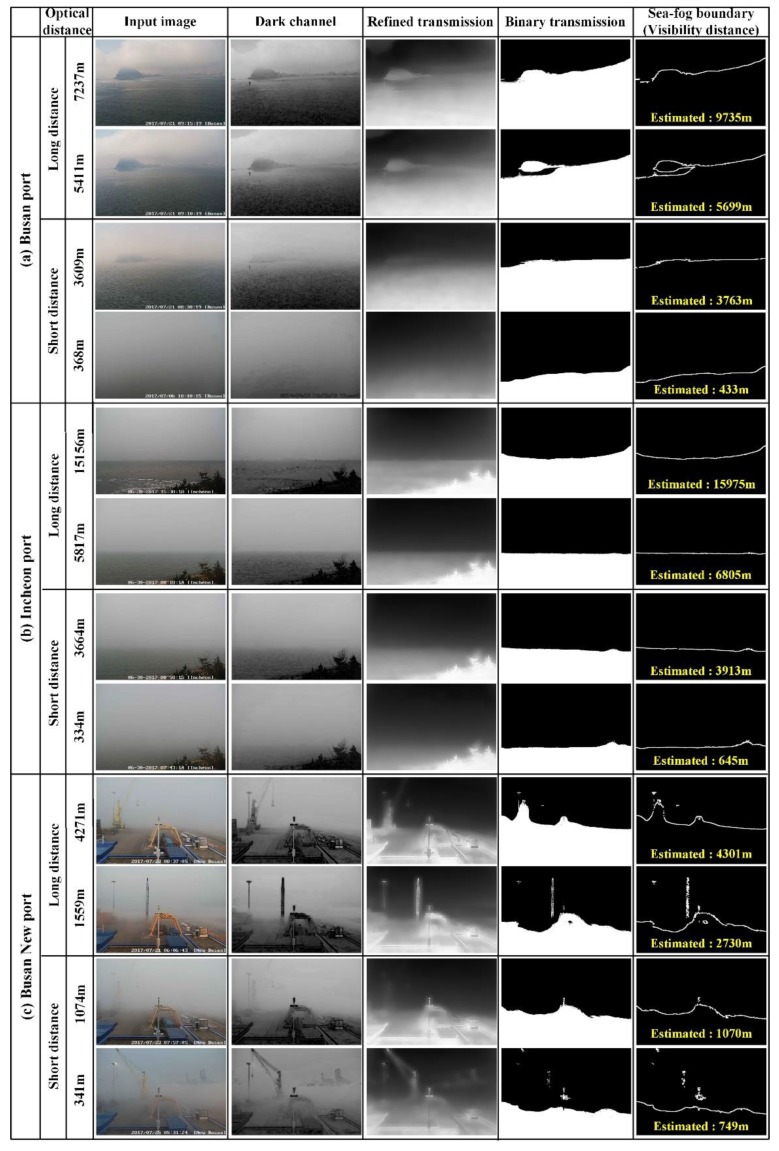
Sea-fog boundary detection and visibility estimation result by proposed method for sea-fog image at (**a**) Pusan port, (**b**) Incheon port, and (**c**) Busan New port.

**Figure 7 sensors-19-04432-f007:**
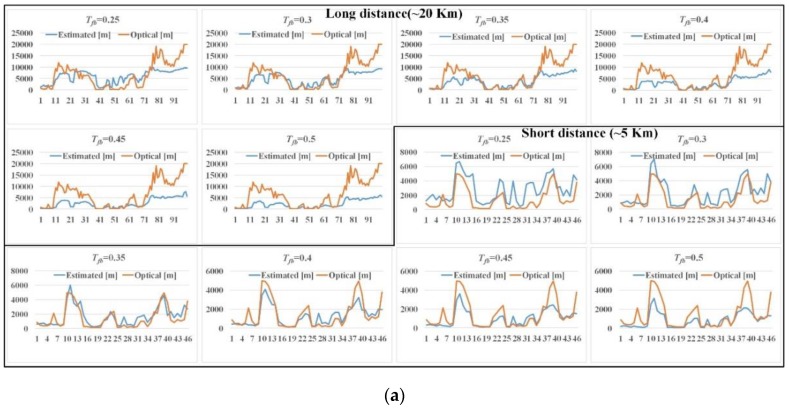
Comparison of estimated visibility distance and optical visibility distance for (**a**) Busan port, (**b**) Incheon port, (**c**) Busan New port according to various thresholds applied to transmission map.

**Figure 8 sensors-19-04432-f008:**
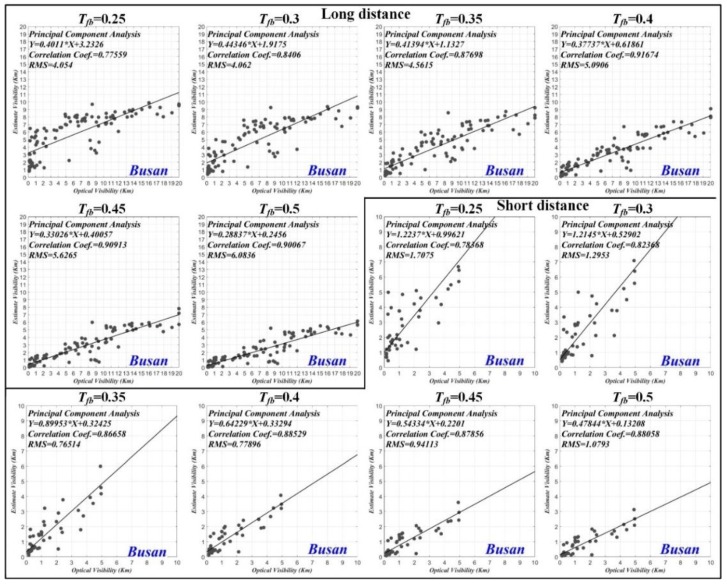
Correlation coefficient result of estimated visibility distance and optical visibility distance for Busan port according to various thresholds applied to transmission map.

**Figure 9 sensors-19-04432-f009:**
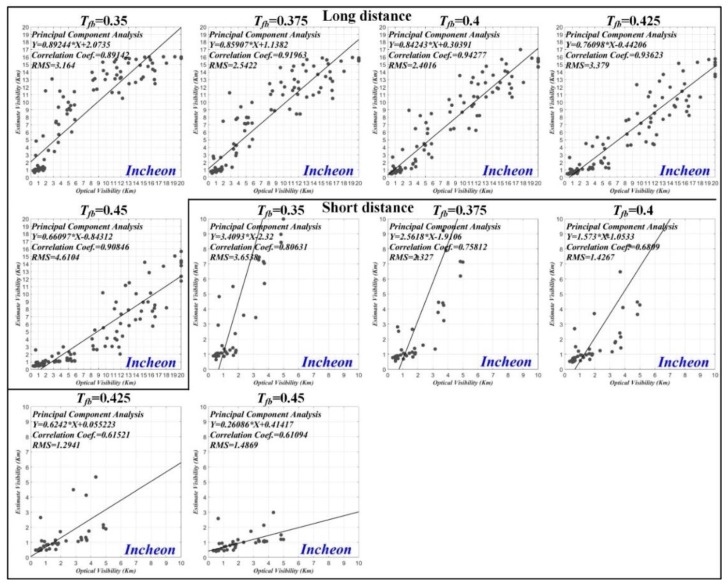
Correlation coefficient result of estimated visibility distance and optical visibility distance for Incheon port according to various thresholds applied to transmission map.

**Figure 10 sensors-19-04432-f010:**
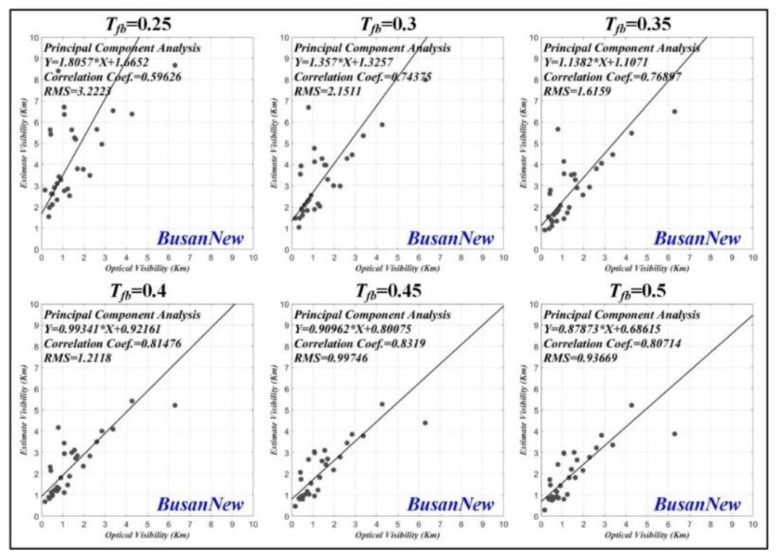
Correlation coefficient result of estimated visibility distance and optical visibility distance for Busan New port according to various thresholds applied to transmission map.

**Figure 11 sensors-19-04432-f011:**
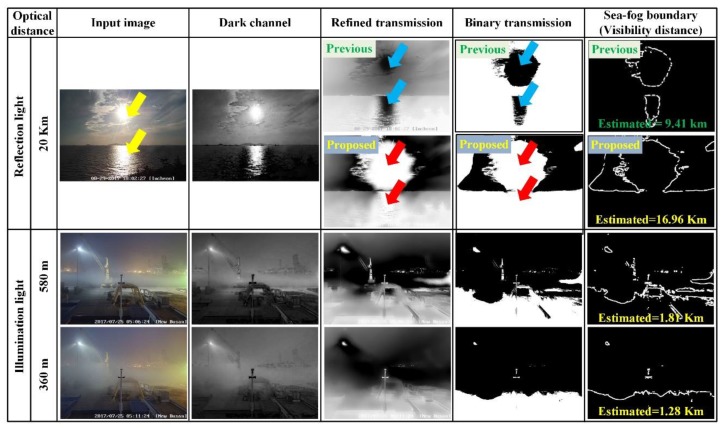
Sea-fog boundary detection and visibility estimation result for abnormal sea-fog image.

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
