# Peer review of "Coastal Visibility Distance Estimation Using Dark Channel Prior and Distance Map Under Sea-Fog: Korean Peninsula Case"

_sensors, 2019, doi:10.3390/s19204432_

Round 1

Reviewer 1 Report

This paper proposes a visibility distance detection algorithm using dark channel prior (DCP) and distance map based on camera image, to improve the local limit of optical visibility sensor and detect the visibility distance of wide area. Experimental results demonstrate the performance of the proposed algorithm. Overall, the proposed method shows a potential application in weather scientists, marine experts, and weather forecasters.

What the role of camera calibration in the framework? Since camera calibration is not available in most cases for customs, so I want to know the effect of this process.

How the distance map influence the accuracy of the proposed algorithm?

Some single image dehazing work can be referred to in the manuscript.

Joint Transmission Map Estimation and Dehazing using Deep Networks, TCSVT 2019

and

Gated Fusion Network for Single Image Dehazing, in CVPR 2018

Author Response

This paper proposes a visibility distance detection algorithm using dark channel prior (DCP) and distance map based on camera image, to improve the local limit of optical visibility sensor and detect the visibility distance of wide area. Experimental results demonstrate the performance of the proposed algorithm. Overall, the proposed method shows a potential application in weather scientists, marine experts, and weather forecasters.

What the role of camera calibration in the framework? Since camera calibration is not available in most cases for customs, so I want to know the effect of this process.

Ans) Yes, right. We performed camera calibration to correct a distortion of the installed CCTV image, but there was no significant difference between the original image and the calibrated image. So the relevant explanations were removed in section 2.3 and the camera calibration part of Figure 1 was removed.

To remove redundancy, the existing Figures 3 and 4 have been merged into the new Figure 3. Figure 5 and 6 were removed. Figure 10-12 were merged into new Figure 6. And existing Figure 9 and 19 were merged into Figure 11.

How the distance map influence the accuracy of the proposed algorithm?

Ans) The distance map explanation of the section 2.3.3 was updated and the following explanation is added in the section 2.3.3.

(Line 195-199) Since the proposed method estimates the visibility distance using distance values corresponding to detected sea-fog boundary, the accuracy of the distance map affects the visibility estimation. The distance map used in the experiments may include a small distance error due to the interpolation because entire distance values were interpolated by some actual distance values. However, we confirmed that it works well in the visibility distance estimation experiment.

Some single image dehazing work can be referred to in the manuscript. Joint Transmission Map Estimation and Dehazing using Deep Networks, TCSVT 2019 And Gated Fusion Network for Single Image Dehazing, in CVPR 2018

Ans) The following explanation is added in the paper.

(Line 86-92) Recently, DCP applications have been developed to reconstruct underwater scenes or hazy images [19-21]. Akkaynak et al [19] proposed a method for removing water from underwater images by estimating the range-dependent attenuation coefficient. Ren et al [20] presented a gated fusion network using three derived input images from an original image and symmetric encoder and decoder for single image dehazing. Zhang et al [21] proposed a unified dehazing network for estimating transmission map precisely. These methods perform very well but have limitations that cannot be applied to heavy fog images that seriously interfere with atmospheric light.

Akkaynak, D.; Treibitz, T. Sea-thru: a method for removing water from underwater images, Proceedings of IEEE Computer Society Conference on Computer Vision and Pattern Recognition (CVPR), CA, USA,16-20, June 2019. Ren, W.; Ma, L.; Zhang, J.; Pan, J.; Cao, X.; Liu, W.; Yang, M.H. Gated fusion network for single image dehazing, Proceedings of IEEE Computer Society Conference on Computer Vision and Pattern Recognition (CVPR), Salt Lake City, USA, 18-22, June 2018, DOI:10.1109/cvpr.2018.00343. Zhang, H.; Sindagi, V.; Patel,V.M. Joint transmission map estimation and dehazing using deep networks, IEEE Transactions on Circuits and Systems for Video Technology 2019, 99, 1-12, DOI: 10.1109/TCSVT.2019.2912145.

Thank you for your comments !!

Reviewer 2 Report

Review of: Coastal Visibility Distance Estimation Using Dark Channel Prior and Distance Map Under Sea-fog Situation: Korean Peninsula Case

Authors propose a visibility distance detection algorithm using dark channel prior (DCP) and distance map based on camera image, to improve the local limit of optical visibility sensor and detect the visibility distance of wide area more accurately.

First, the dark channel for an input sea-fog image is calculated. Then, a binary transmission image is obtained by applying a threshold to the estimated transmission obtained from the dark channel and air-light.

Then, the sum of distance values of pixels corresponding to fog boundary extracted by simple image processing is averaged to derive final visibility distance.

The paper also proposes a new air-light and transmission estimation method in order to extract a visibility distance from an abnormal sea-fog image including sunlight and illumination light.

**************************

General comment that applies throughout the manuscript: too long! Sentences are very long, paragraphs are very long, too much background information, and there is a lot of repetition. A lot of content can be cut/rephrased. As is, it is very long, makes it hard for me to get to the meat to do the review, and will likely deter readers too. Please consider rewriting so that the length is at least half! Currently, it is 22 pages! This does not mean you will have to omit important details. On the contrary, when you write more concise, the important details will be easy to come across. Consider putting some of the results/methods into Supplementary material. For example, Fig 6: totally unnecessary. Users should be able to understand what a binary mask is, and what Eq 8 says, so no need for you to show redundant information using Excel sheets. Move some of the correlations to Supplementary, only report what's critical in text. Or cut.

*************************

What is termed "Image Degradation Model" is actually called "Image Formation Model".

Need a reference for this as there can be varying definitions:

Fog is defined as a phenomenon in which very small droplets float in the atmosphere and the horizontal visibility is less than 1 Km.

Need reference:
The extinction coefficient can also be named fog density.

What does this mean: "straightness of light" ?

"ensure visibility" is not meaningful. perhaps consider "improve visibility"?

"Sea-fog means the fog that occurs in the sea". Is it IN the sea, or over the sea surface?

It's not clear to me how the distance  map is obtained. This is the explanation: "Therefore, the distance map is generated by converting the vertical 241 pixel distance value from the lowest pixel to each object (islands) pixel in the camera image into the 242 actual distance from the CCTV installation position to each object. "

This does not seem to be a true distance map. Only a single point on the map is known and the rest are simply interpolated? Otherwise objects at different ranges would show up in the distance map, rather than straight lines. How is the actual distance measured? Satellites?

If you have a true distance map, you can practically remove the fog from your images, please see Sea-thru: A Method For Removing Water from Underwater Images, Akkaynak & Treibitz, CVPR 2019. 

Line 300: solve these drawbacks --> address these drawbacks. Generally, please have a native English speaker read over the manuscript for minor issues like this.

Overall, I like the method. It is very simple -- but that does not mean it's bad. And it addresses an important problem. However, the writing and presentation has to improve, because in its current format, it is hard to understand the contributions.

Author Response

Authors propose a visibility distance detection algorithm using dark channel prior (DCP) and distance map based on camera image, to improve the local limit of optical visibility sensor and detect the visibility distance of wide area more accurately. First, the dark channel for an input sea-fog image is calculated. Then, a binary transmission image is obtained by applying a threshold to the estimated transmission obtained from the dark channel and air-light. Then, the sum of distance values of pixels corresponding to fog boundary extracted by simple image processing is averaged to derive final visibility distance. The paper also proposes a new air-light and transmission estimation method in order to extract a visibility distance from an abnormal sea-fog image including sunlight and illumination light.

General comment that applies throughout the manuscript: too long! Sentences are very long, paragraphs are very long, too much background information, and there is a lot of repetition. A lot of content can be cut/rephrased. As is, it is very long, makes it hard for me to get to the meat to do the review, and will likely deter readers too. Please consider rewriting so that the length is at least half! Currently, it is 22 pages! This does not mean you will have to omit important details. On the contrary, when you write more concise, the important details will be easy to come across. Consider putting some of the results/methods into Supplementary material. For example, Fig 6: totally unnecessary. Users should be able to understand what a binary mask is, and what Eq 8 says, so no need for you to show redundant information using Excel sheets. Move some of the correlations to Supplementary, only report what's critical in text. Or cut.

Ans) Yes. I agree. Figure 6 has been removed. And the original paper of 22 pages was reduced to 15 pages according to your advice.

In addition, we have updated the figures according to your advice. To remove redundancy, the existing Figures 3 and 4 have been merged into the new Figure 3. Figure 5 and 6 were removed. Figure 10-12 were merged into new Figure 6. And existing Figure 9 and 19 were merged into Figure 11. Also, the descriptions of the figures were updated.

What is termed "Image Degradation Model" is actually called "Image Formation Model".

Ans) Right.

(Line 123) We replaced "Image Degradation Model" with "Image Formation Model" with the following reference.

Fattal, R. Single image dehazing, ACM Trans. Graph. 2008, 72, 1-9, DOI: 10.1145/1360612.1360671.

Need a reference for this as there can be varying definitions:

Fog is defined as a phenomenon in which very small droplets float in the atmosphere and the horizontal visibility is less than 1 Km.

Ans) Yes, right. We added the following reference for the fog explanation.

(Line 60) 3. Gultepe I.; Milbrandt, J.A.; Zhou,B. Marine fog: a review on microphysics and visibility prediction, Springer International Publishing: Cham, Switzerland, 2017; pp. 345-394.

Need reference:

The extinction coefficient can also be named fog density.

Ans) We added the following reference for the sentence.

(Line 67) 5. Azzawi, M.M.; Maliek, H.A. Optical forward scattering property employment for measuring the atmospheric visibility, J. of College of Education for Women 2016, 27, 1485-1492.

What does this mean: "straightness of light" ?

Ans) The verse has been changed as follows.

(Line 69-70) due to the nature of light going straight

"ensure visibility" is not meaningful. perhaps consider "improve visibility"?

Ans) Yes, right.

(Line 72) We revised the word.

"Sea-fog means the fog that occurs in the sea". Is it IN the sea, or over the sea surface?

Ans) Yes, sea surface is right. Since the explanation follows one after another, the sentence is deleted.

(Line 108) “Sea surface: term was used for the explanation.

It's not clear to me how the distance map is obtained. This is the explanation: "Therefore, the distance map is generated by converting the vertical 241 pixel distance value from the lowest pixel to each object (islands) pixel in the camera image into the 242 actual distance from the CCTV installation position to each object. "

This does not seem to be a true distance map. Only a single point on the map is known and the rest are simply interpolated? Otherwise objects at different ranges would show up in the distance map, rather than straight lines. How is the actual distance measured? Satellites?

Ans) The sentence was briefly changed as the following.

(Line 265) The distance map is obtained by interpolating actual distance value to each island and some actual distances obtained from Google satellite map.

And we revised section 2.3.3 (Distance Map) as the following.

(Line 187-199) Figure 3 also shows the distance map for each port used to calculate the visibility distance in the proposed visibility distance detection method. The distance map is generated by interpolating with actual distance from each CCTV position to each island and some obtained actual distances (the blue dot in the figure of the fourth column) for the vertical pixels of the images. In the figure, the second column shows actual distance from each CCTV position to each island. And the third and fourth column respectively show the estimated distance map and the distance values for vertical pixels in the estimated distance map. In the figures of the fourth column, the blue dot represents actual distance values obtained from Google satellite map, and the red line represents the interpolated distance values. Since the proposed method estimates the visibility distance using distance values corresponding to detected sea-fog boundary, the accuracy of the distance map affects the visibility estimation. The distance map used in the experiments may include a small distance error due to the interpolation because entire distance values were interpolated by some actual distance values. However, we confirmed that it works well in the visibility distance estimation experiment.

If you have a true distance map, you can practically remove the fog from your images, please see Sea-thru: A Method For Removing Water from Underwater Images, Akkaynak & Treibitz, CVPR 2019.

Ans) A description of the paper that informed us and additional papers was added as the following.

(Line 86-92) Recently, DCP applications have been developed to reconstruct underwater scenes or hazy images [19-21]. Akkaynak et al [19] proposed a method for removing water from underwater images by estimating the range-dependent attenuation coefficient. Ren et al [20] presented a gated fusion network using three derived input images from an original image and symmetric encoder and decoder for single image dehazing. Zhang et al [21] proposed a unified dehazing network for estimating transmission map precisely. These methods perform very well but have limitations that cannot be applied to heavy fog images that seriously interfere with atmospheric light.

Akkaynak, D.; Treibitz, T. Sea-thru: a method for removing water from underwater images, Proceedings of IEEE Computer Society Conference on Computer Vision and Pattern Recognition (CVPR), CA, USA,16-20, June 2019. Ren, W.; Ma, L.; Zhang, J.; Pan, J.; Cao, X.; Liu, W.; Yang, M.H. Gated fusion network for single image dehazing, Proceedings of IEEE Computer Society Conference on Computer Vision and Pattern Recognition (CVPR), Salt Lake City, USA, 18-22, June 2018, DOI:10.1109/cvpr.2018.00343. Zhang, H.; Sindagi, V.; Patel,V.M. Joint transmission map estimation and dehazing using deep networks, IEEE Transactions on Circuits and Systems for Video Technology 2019, 99, 1-12, DOI: 10.1109/TCSVT.2019.2912145.

Line 300: solve these drawbacks --> address these drawbacks. Generally, please have a native English speaker read over the manuscript for minor issues like this.

Ans) Yes, right. (Line 234) The word was revised according to your advice.

Overall, I like the method. It is very simple -- but that does not mean it's bad. And it addresses an important problem. However, the writing and presentation has to improve, because in its current format, it is hard to understand the contributions.

Ans) Thank you for the advice. The writing has been checked throughout the paper.

Thank you for your comments !!

Round 2

Reviewer 2 Report

This is my second review of the manuscript titled: “Coastal Visibility Distance Estimation Using Dark Channel Prior and Distance Map Under Sea-fog Situation: Korean Peninsula Case”. The authors have quickly provided an updated manuscript, answering each of my comments. I have read the updated manuscript in detail, and below are my final comments (mostly minor and aimed towards further clarification):

I would remove "situation" from title

Line 69:

I am still struggling with what this means in terms of physics:

However, due to the nature of light going straight, it is still very difficult to generate comprehensive visual information for a large area such as the coast.

Light may travel on a straight line, but also scatters in all directions, so from that aspect, it may “turn corners”. Where exactly is the difficulty in mapping large coastal areas coming from? Could you describe in a bit more detail providing the physics?

Line 85: Actually DCP does not work well underwater at all, so please reword the following paragraph (underlines are my suggestions, but edit as needed):

Recently, DCP applications have been developed to reconstruct underwater scenes or hazy images [19-21], although its direct application to underwater images as-is is not suitable and creates inconsistent results. Akkaynak et al [19] modified the DCP method for underwater images and proposed a method for removing water from underwater images by estimating the range-dependent attenuation coefficient. Ren et al [20] presented a gated fusion network using three derived input images from an original image and symmetric encoder and decoder for single image dehazing. Zhang et al [21] proposed a unified dehazing network for estimating transmission map precisely (a bit more detail here because it is not clear why this work is being mentioned).

Line 90:

These methods perform very well but have limitations that cannot be applied to heavy fog images that seriously interfere with atmospheric light.

Have you actually applied them to your scenes? If not, I would phrase this sentence as “we expect they will have limitations”, and also, I would list where/what you expect limitations to be. For example, I have extensive experience in underwater scene reconstruction, and I disagree with your statement. Correcting underwater scenes is much more difficult than atmospheric scenes, because the physics are much complex, and even tough the distances are smaller, the light field is more complex. If a physics-based method works well on an underwater scene, I would expect it to work well in an atmospheric scene. But the key point here is “physics-based”. For example underwater dehazing methods were failing for the last 30 years because they were using an equation with incorrect physics—an atmospheric equation for underwater scenes (see Akkaynak & Treibitz 2018: A Revised Underwater Image Formation model). Once the physics were corrected, correcting colors in underwater images became an almost-solved problem (Akkaynak & Treibitz 2019). I would expect that their Sea-thru method (which requires a known distance map) would work wonderfully on your images, so without trying, I would not automatically dismiss that they cannot work on heavy fog images.

A potential area for you to look (and cite if appropriate) is the atmospheric correction(s) that is done for remotely sensed ocean color data. Those methods involve removing clouds, which is a way of dehazing, and often use machine learning to do so. Many of these algorithms are very robust and commonly used by satellite imaging communities. I am not immediately sure they are directly relevant to your work, but for the sake of completeness of surveying dehazing/water removal methods, I would suggest quickly reviewing them.

Line 94:

Calculated -- > formed

Line 103:

Types (add “s)

Line 122: Please add that this equation is for a hazy image in the atmosphere only, since “dehazing” is also often used for underwater images and the equation is different for underwater (see my comment above about the new image formation model for underwater by Akkaynak & Treibitz, which is different than atmospheric model). Also, if you read the PhD thesis of He Kaiming, you will see that he also proposes a model similar to Akkaynak & Treibitz, with 2 different coefficients, for heavy atmospheric conditions. So your work might benefit from the Akkaynak – Treibitz model, instead of Eq. 2 on line 122.

Methods

The overall writing is much more clear now, and I was able to follow the details as if I was going to implement them. It was well worth your effort shortening and consolidating the figures. Thank you.

Line 286

I still recommend having the manuscript be reviewed by a native English speaker, not for just typos or spellchecking, but for expressions – which will benefit the clarity of your writing, and will promote more citations of your paper. For example, this sentence would not be written as such by a native English speaker:

“We can know that the estimated visibility distance is very similar to the optical visibility distance”

or the opening sentence in Introduction:

So far, weather scientists, marine experts, and weather forecasters who study marine weather have studied various marine weather conditions including sea-fog, typhoons, and high seas (or high winds).

Line 361

Fig 10 is a bit small to read like this, you can make it bigger.

Line 389

What do you mean by “locality”?

Line 404 – remove “are”

Author Response

This is my second review of the manuscript titled: “Coastal Visibility Distance Estimation Using Dark Channel Prior and Distance Map Under Sea-fog Situation: Korean Peninsula Case”. The authors have quickly provided an updated manuscript, answering each of my comments. I have read the updated manuscript in detail, and below are my final comments (mostly minor and aimed towards further clarification): I would remove "situation" from title Ans) Yes, that would be better. It was removed. Line 69: I am still struggling with what this means in terms of physics: However, due to the nature of light going straight, it is still very difficult to generate comprehensive visual information for a large area such as the coast. Light may travel on a straight line, but also scatters in all directions, so from that aspect, it may “turn corners”. Where exactly is the difficulty in mapping large coastal areas coming from? Could you describe in a bit more detail providing the physics? Ans) The explanation was ambiguous. The sentence was revised as follows: However, it is very expensive to deploy an optical visibility sensor to obtain comprehensive visibility information for large areas such as the coast. Line 85: Actually DCP does not work well underwater at all, so please reword the following paragraph (underlines are my suggestions, but edit as needed): Recently, DCP applications have been developed to reconstruct underwater scenes or hazy images [19-21], although its direct application to underwater images as-is is not suitable and creates inconsistent results. Akkaynak et al [19] modified the DCP method for underwater images and proposed a method for removing water from underwater images by estimating the range-dependent attenuation coefficient. Ren et al [20] presented a gated fusion network using three derived input images from an original image and symmetric encoder and decoder for single image dehazing. Zhang et al [21] proposed a unified dehazing network for estimating transmission map precisely (a bit more detail here because it is not clear why this work is being mentioned). Ans) Thank you for your kind suggestion. We made that correction. In addition, the reason of the introduced method is added as below. These introduced methods can be used in vehicles and ships to ensure visibility in coastal areas. Line 90: These methods perform very well but have limitations that cannot be applied to heavy fog images that seriously interfere with atmospheric light. Have you actually applied them to your scenes? If not, I would phrase this sentence as “we expect they will have limitations”, and also, I would list where/what you expect limitations to be. For example, I have extensive experience in underwater scene reconstruction, and I disagree with your statement. Correcting underwater scenes is much more difficult than atmospheric scenes, because the physics are much complex, and even tough the distances are smaller, the light field is more complex. If a physics-based method works well on an underwater scene, I would expect it to work well in an atmospheric scene. But the key point here is “physics-based”. For example underwater dehazing methods were failing for the last 30 years because they were using an equation with incorrect physics—an atmospheric equation for underwater scenes (see Akkaynak & Treibitz 2018: A Revised Underwater Image Formation model). Once the physics were corrected, correcting colors in underwater images became an almost-solved problem (Akkaynak & Treibitz 2019). I would expect that their Sea-thru method (which requires a known distance map) would work wonderfully on your images, so without trying, I would not automatically dismiss that they cannot work on heavy fog images. Ans) Thank you for giving me good information. I agree with you and have made the following modifications. These DCP-based findings can also be applied to quantitatively estimate the visibility distance in coastal areas where sea-fog occurs frequently. A potential area for you to look (and cite if appropriate) is the atmospheric correction(s) that is done for remotely sensed ocean color data. Those methods involve removing clouds, which is a way of dehazing, and often use machine learning to do so. Many of these algorithms are very robust and commonly used by satellite imaging communities. I am not immediately sure they are directly relevant to your work, but for the sake of completeness of surveying dehazing/water removal methods, I would suggest quickly reviewing them. Ans) Thank you for the new information on that research area. The following descriptions and references have been added to the discussion section. Additionally, atmospheric correction techniques for distinguishing and water contributions have been researched in the ocean color remote sensing [25, 26]. The studies are related to the absorption or reflection properties of water in the infrared wavelength band, modeling water properties, and ocean inversion using neural networks. These studies can also provide global atmospheric and ocean segmentation information in coastal areas. Therefore, if the proposed technique is combined with the atmospheric correction technique in the future, more reliable visibility distance may be estimated. 25. Bertrand S.; Ronan F.; Ludovic B.; Grégoire M.; Odile F.A. MEETC2: Ocean color atmospheric corrections in coastal complex waters using a Bayesian latent class model and potential for the incoming sentinel 3 - OLCI mission, Remote Sensing of Environment 2016, 172, 39–49, DOI: ff10.1016/j.rse.2015.10.035ff. ffhal-01611644f. 26. Mohamed A.M.; Cédric J.; Hubert L.; Vincent V.; Xavier M.; Arnaud C. Evaluation of five atmospheric correction algorithms over french optically-complex waters for the sentinel-3A OLCI ocean color sensor, Sensors 2019, 11, 1–25, https://doi.org/10.3390/rs11060668. Line 94: Calculated -- > formed Ans) I revised it. Line 103: Types (add “s) Ans) I revised it. Line 122: Please add that this equation is for a hazy image in the atmosphere only, since “dehazing” is also often used for underwater images and the equation is different for underwater (see my comment above about the new image formation model for underwater by Akkaynak & Treibitz, which is different than atmospheric model). Also, if you read the PhD thesis of He Kaiming, you will see that he also proposes a model similar to Akkaynak & Treibitz, with 2 different coefficients, for heavy atmospheric conditions. So your work might benefit from the Akkaynak – Treibitz model, instead of Eq. 2 on line 122. Ans) Based on your opinion, the sentence was revised as follows: The haze imaging equation [9] for a hazy image in the atmosphere is given as follows: Methods The overall writing is much more clear now, and I was able to follow the details as if I was going to implement them. It was well worth your effort shortening and consolidating the figures. Thank you. Ans) Thank you. Line 286 I still recommend having the manuscript be reviewed by a native English speaker, not for just typos or spellchecking, but for expressions – which will benefit the clarity of your writing, and will promote more citations of your paper. For example, this sentence would not be written as such by a native English speaker: “We can know that the estimated visibility distance is very similar to the optical visibility distance” or the opening sentence in Introduction: So far, weather scientists, marine experts, and weather forecasters who study marine weather have studied various marine weather conditions including sea-fog, typhoons, and high seas (or high winds). Ans) The sentence was corrected as the following. We also checked the paper again. It is experimentally demonstrated that the estimated visibility distance is very similar to the optical visibility distance. Detection for various marine weathers such as sea-fog, typhoons, and high seas (or high winds) is being studied for the safety of civil and marine transit. Line 361 Fig 10 is a bit small to read like this, you can make it bigger. Ans) The figure was enlarged. Line 389 What do you mean by “locality”? Ans) The vague description has been corrected as follows: To replace the existing optical visibility sensor and obtain a comprehensive visibility distance information in the coastal area, Line 404 – remove “are” Ans) I revised it. Thank you for the advice !!
